

# Association between genetic polymorphisms of the *IL28B* gene and leukomonocyte in Chinese hepatitis B virus-infected individuals

Yuzhu Song[1,*], Yunsong Shen[2,*], Xueshan Xia[1] and A-Mei Zhang[1]

[1] Faculty of Life Science and Technology, Kunming University of Science and Technology, Kunming, Yunnan, China
[2] First People's Hospital of Yunnan Province, Kunming, Yunnan, China
[*] These authors contributed equally to this work.

Corresponding author
A-Mei Zhang, zam1980@yeah.net

## ABSTRACT

**Background**. Hepatitis B infection is one of the most severe hepatic diseases in China. Thus, understanding the genetic pathogenesis of hepatitis B virus (HBV)-infected individuals is important. Although no consistent result is obtained in different populations, HBV treatment effect is reportedly associated with the *IL28B* gene.

**Methods**. To investigate the role of the *IL28B* gene in HBV-infected individuals in Yunnan, China, we screened genotypes of three single nucleotide polymorphisms (SNPs, rs12979860, rs8099917, and rs12980275) in HBV-infected individuals and general controls by using SnapShot and sequencing.

**Results**. Results showed no significant difference was found in genotypes, alleles, and haplotypes frequency between the HBV-infected individuals and controls. After dividing the HBV-infected individuals into patients in acute infection, chronic HBV patients, and patients undergoing convalescence, the genotype GT ($P = 0.033$) and allele G ($P = 0.038$) of rs8099917 showed statistically higher frequency in the acutely infectious individuals than in the HBV patients undergoing convalescence. HBV viral load was higher in the acutely infectious patients than in the chronic infection group. Strikingly, we found that leukomonocyte (LYM) level was associated with SNPs in the *IL28B* gene. In addition, the LYM levels were lower in the HBV-infected individuals with genotype CC of rs12979860 and AA of rs12980275 than in the patients with other genotypes of these two SNPs.

**Conclusion**. Our results suggested genetic polymorphisms of the *IL28B* gene were associated with LYM level of HBV-infected individuals.

## INTRODUCTION

Although hepatitis B virus (HBV) vaccine is widely used, HBV infection remains a main cause of hepatic cirrhosis and hepatocellular carcinoma (*EASL, 2012*). Among 2 billion people infected with HBV, 350 million individuals develop chronic HBV infection (*Liaw & Chu, 2009*). Over 8% of Chinese, Southeast Asian, and African populations are hepatitis

B surface antigen (HBsAg)-positive (*Trepo, Chan & Lok, 2014*), so HBV infection is still a serious problem in these populations. Genetic factors influence HBV-infection and treatment effects of HBV-infected individuals (*Hu et al., 2012*; *Li et al., 2012*; *Wasityastuti et al., 2016*). Understanding the role of host genetic variations in HBV infection, pathogenesis, and therapy is important to provide individualized protection and treatment for HBV-infected cases.

Interleukin 28B (IL28B) belongs to the interferon λ family, which is a new interferon family. Genetic variations of the *IL28B* gene have been identified to be associated with HCV infection, viral clearance, and response to therapy (*Suppiah et al., 2009*; *Tanaka et al., 2009*; *Thomas et al., 2009*; *Zhang et al., 2014*). However, genome-wide association studies (GWAS) in Asian population found that *IL28B* gene is not correlated with HBV infection or viral clearance (*Kamatani et al., 2009*; *Hu et al., 2013*). *Kamatani et al. (2009)* identified that genetic variants in the HLA-DP locus are strongly associated with risk of persistent HBV infection. Subsequently, two new gene loci (HLA-C and UBE2L3) have been highlighted to play important roles in the clearance of HBV infection (*Hu et al., 2013*). However, the single nucleotide polymorphisms (SNPs) or the haplotypes constructed by SNPs in the *IL28B* gene can influence the HBV infection, HBV surface antigen seroclearance, or treatment of HBV-infected individuals in special cohorts (*Seto et al., 2013*; *Domagalski et al., 2014*). SNPs rs12979860 and rs8099917 were mostly studied and identified to be associated with HBV-infection in Chinese (*Chen et al., 2015*). Until now, we found no study was performed to reveal the relationship of genetic polyporphisms in the *IL28B* of HBV-infected persons and biochemical characteristics. The purpose of this study is to investigate whether SNPs in the *IL28B* gene influence the HBV infection and biochemical characteristics of HBV-infected individuals in Yunnan, China.

## MATERIALS AND METHODS

### Subjects

All HBV-infected individuals and age- and gender-matched controls were recruited by doctors in the First People's Hospital of Yunnan Province. In our previous study, basic information, biochemical characteristics, and HBV serological markers of each subject were reported (*Song et al., 2016*). In brief, the liver function test (alanine transaminease (ALT), aspartate transaminase (AST), total bilirubin (TBIL), direct bilirubin (DBIL), indirect bilirubin (IBIL), total protein (TP), albumin (ALB), and globin (GLOB)), renal function (blood urea nitrogen (BUN), serum creatinine (CREA), serum uric acid (UA), and blood glucose (GLU)), and part of blood routine (white blood cells (WBC), neutrophilic granulocyte (NEUT), lymphocytes (LYM), monocytes (MONO), eosinophil granulocyte (EO), and basophile granulocyte (BASO)) were measured (all data has been reported in our previous study (*Song et al., 2016*)). All individuals were Han Chinese. In this study, whole blood samples (3 mL) of 493 HBV-infected individuals (274 males and 219 females) and 460 controls (285 males and 175 females) were collected. All HBV-infected individuals and controls were not infected with hepatitis C virus (HCV), human immunodeficiency virus (HIV), and *Treponema pallidum*. All controls were devoid of HBV, HCV, or HIV infection,
and no seriously hepatitic disease (hepatic fibrosis, Hepatocellular Carcinoma, and so forth) were identified in controls. The mean ages of the individuals in the HBV-infected and control cohorts were 41.5 ± 0.3 and 38.8 ± 0.3 years (mean ± SEM), respectively. HBV-infected individuals were also without other serious liver disease. Written informed consent conforming to the tenets of the Declaration of Helsinki was obtained from each participant prior to the study. This study was approved by the Institutional Review Board of Kunming University of Science and Technology (Approval No. 2014SK027).

## Genomic DNA extraction and genotyping

Genomic DNA was extracted from whole blood by using the TIANamp genomic DNA Kit (TIANGEN, China). Three SNPs (i.e., rs12979860, rs8099917, and rs12980275) in the *IL28B* gene were selected and analyzed as reference-described (*Chen et al., 2015*). Genotypes of each SNP were screened by using the SnapShot assay (Table S1). SnapShot is based on single-base extension. In brief, amplifying and extending primers for each SNP are firstly designed. Then PCR reaction is performed with two kinds of primers and will be terminated behind one base of 3′ end of extending primers. Finally, genotypes of each SNP are determined according to fluorescence color. Genotyping results of 10% total samples were verified by sequencing. Haplotypes were constructed using Phase software for further analysis (*Stephens & Donnelly, 2003*).

## Subgrouping of HBV-infected individuals

According to the subgrouping principle in our previous study (*Song et al., 2016*), HBV-infected individuals were divided into three groups. Group #1 included patients who with HbsAg and HBeAg positive were at acute infected phase ($N = 29$). Group #2 included patients with HBsAg and anti-HBc positive, who were at chronic infected phase ($N = 202$). Patients in group #3 were comprised of HBV-infected individuals undergoing convalescence ($N = 262$), who were anti-HBc positive but HBV DNA negative.

## Quantification of the HBV viral load

HBV DNA was extracted from the serum of each sample by using the TIANamp Virus DNA/RNA Kit (TIANGEN, China). The HBV viral load in the serum of 58 HBV-infected individuals was detected by using the Hepatitis B Viral DNA Quantitative Fluorescence Diagnostic Kit (realtime-PCR-Fluorescence Probing) (Sansure, Hunan, China) in accordance with the manufacturer's instructions and ABI 7500 Fast Real-Time PCR system (Applied Biosystems, USA). Results were recorded in International Unit (IU)/mL; one IU/mL was equivalent to 5.6 copies/mL. The lowest detection limitation of this quantitative method was 2,800 copies/mL (500 IU/mL). In further analysis, HBV DNA was log10-transformed, i.e., the lowest detection limitation was 3.45 log10 copies/mL.

## Data analysis

The Hardy–Weinberg equilibrium (HWE) was assessed in each variant to evaluate the deviation of collected cohorts. Given that the numbers of a genotype of each SNP were less than five, 1,000 iterations for simulation were performed. The Chi-square test was collected to analyze the frequency of genotypes and alleles between the different cohorts.

**Table 1  Genotype and allele frequency of three SNPs of the *IL28B* gene in HBV-infected individuals and controls.**

| SNP (genotype/allele) | | HBV-infected individual ($N = 493$) | Controls ($N = 460$) | *P*-value[b] | OR | 95% CI |
|---|---|---|---|---|---|---|
| rs12979860 | | HWE[a] $P = 1.00$ | HWE $P = 0.79$ | | | |
| | CC | 439 | 400 | 0.320 | 1.219 | 0.824–1.804 |
| Genotype | CT | 52 | 57 | 0.372 | 0.834 | 0.559–1.243 |
| | TT | 2 | 3 | 0.599 | 0.621 | 0.103–3.730 |
| Allele | C | 930 | 857 | 0.292 | 1.221 | 0.842–1.771 |
| | T | 56 | 63 | | 0.819 | 0.565–1.188 |
| rs8099917 | | HWE $P = 1.00$ | HWE $P = 1.00$ | | | |
| | GG | 1 | 2 | 0.523 | 0.465 | 0.042–5.151 |
| Genotype | GT | 49 | 52 | 0.494 | 0.866 | 0.573–1.308 |
| | TT | 443 | 406 | 0.429 | 1.178 | 0.784–1.771 |
| Allele | G | 51 | 56 | 0.386 | 0.842 | 0.570–1.244 |
| | T | 935 | 864 | | 1.188 | 0.804–1.756 |
| rs12980275 | | HWE $P = 0.78$ | HWE $P = 0.50$ | | | |
| | AA | 437 | 398 | 0.321 | 1.216 | 0.826–1.788 |
| Genotype | AG | 55 | 58 | 0.488 | 0.870 | 0.588–1.298 |
| | GG | 1 | 4 | 0.155 | 0.232 | 0.026–2.081 |
| Allele | A | 929 | 854 | 0.216 | 1.260 | 0.873–1.817 |
| | G | 57 | 66 | | 0.794 | 0.550–1.145 |

Notes.

[a] Chi-square test for deviation from the Hardy–Weinberg equilibrium (a value of $P < 0.01$ was regarded as a deviation from the HWE).

[b] Chi-square test was used.

Student's $t$-test (unpaired, two tails) was used to compare HBV viral load between the two HBV-infected groups and biochemical characteristics among the HBV-infected individuals with different genotypes. Biochemical characteristics in the groups are presented as mean $\pm$ SEM. Correlation analysis was used to analyze the relationship between viral loads and LYM levels. Statistical significance was considered at $P < 0.05$.

# RESULTS

No deviation was found in the HBV-infected individuals and controls after calculating HWE, and these results suggested that the analyzed population was in genetic equilibrium. Genotype and allele frequency showed no significant difference between the HBV-infected individuals and general controls (Table 1). The genotyping data are listed in Tables S2 and S3 (Table S2 for HBV-infected patients and Table S3 for controls). Seven and five haplotypes constructed by three SNPs were identified in the HBV-infection group and controls, respectively. Although the haplotypes were somewhat different between two groups, the frequency of each haplotype showed no significant difference (Table 2).

HBV-infected individuals were divided into three subgroups according to HBV serological markers of patients. Excluding the genotype and allele frequency of SNP rs8099917 between groups #1 and #3, we identified no other significant difference (Table 3). The genotype GT of rs8099917 showed a statistically higher frequency in group #1 (20.69%, 6/29) than in group #3 (8.40%, 22/262) ($P = 0.033$), whereas the genotype TT of rs8099917

**Table 2  Haplotypes constructed by three SNPs in HBV-infected individuals and controls.**

| Haplotype | HBV-infected individuals (N = 493) | Controls (N = 460) | P-value[b] | OR | 95% CI |
|---|---|---|---|---|---|
| CTA | 925 | 853 | 0.339 | 1.191 | 0.832–1.706 |
| TGG | 46 | 55 | 0.201 | 0.770 | 0.515–1.151 |
| TTG | 7 | 7 | 0.896 | 0.933 | 0.326–2.669 |
| Others[a] | 8 | 5 | 0.478 | 1.497 | 0.488–4.592 |

**Notes.**
[a] Means those haplotypes which frequency is less than 0.5%.
[b] Chi-square test was used.

**Table 3  Genotype and allele frequency in three groups of HBV-infected individuals.**

| SNP (genotype/allele) | | Group #1 (N = 29) | Group #2 (N = 202) | Group #3 (N = 262) | Group #1 v.s. group #2[a] P-value (OR, 95% CI)[b] | Group #1 v.s. group #3 P-value (OR, 95% CI) | Group #2 v.s. group #3 P-value (OR, 95% CI) |
|---|---|---|---|---|---|---|---|
| **rs12979860** | | | | | | | |
| | CC | 24 | 178 | 237 | 0.415 (0.647, 0.226–1.856) | 0.196 (0.506, 0.178–1.444) | 0.416 (0.782, 0.432–1.415) |
| Genotype | CT | 5 | 22 | 25 | 0.320 (1.705, 0.590–4.922) | 0.196 (1.975, 0.693–5.632) | 0.633 (1.159, 0.633–2.121) |
| | TT | 0 | 2 | 0 | 0.590 (–) | 1.000 (–) | 0.107 (–) |
| Allele | C | 53 | 378 | 499 | 0.534 (0.729, 0.268–1.981) | 0.208 (0.531, 0.195–1.445) | 0.270 (1.373, 0.780–2.416) |
| | T | 5 | 26 | 25 | 0.534 (1.372, 0.505–3.726) | 0.208 (1.883, 0.692–5.124) | 0.270 (0.728, 0.414–1.282) |
| **rs8099917** | | | | | | | |
| | GG | 0 | 1 | 0 | 0.704 (–) | 1.000 (–) | 0.254 (–) |
| Genotype | GT | 6 | 21 | 22 | 0.107 (2.248, 0.822–6.147) | 0.033 (2.846, 1.048–7.728) | 0.462 (1.266, 0.675–2.372) |
| | TT | 23 | 180 | 240 | 0.131 (0.469, 0.172–1.276) | 0.033 (0.351, 0.129–0.954) | 0.363 (0.750, 0.403–1.397) |
| Allele | G | 6 | 23 | 22 | 0.172 (1.911, 0.744–4.913) | 0.038 (2.633, 1.022–6.786) | 0.293 (1.377, 0.756–2.509) |
| | T | 52 | 381 | 502 | 0.172 (0.523, 0.204–1.345) | 0.038 (0.380, 0.147–0.979) | 0.293 (0.726, 0.399–1.322) |
| **rs12980275** | | | | | | | |
| | AA | 24 | 177 | 236 | 0.466 (0.678, 0.237–1.939) | 0.226 (0.529, 0.186–1.504) | 0.402 (0.780, 0.436–1.397) |
| Genotype | AG | 5 | 24 | 26 | 0.415 (1.545, 0.539–4.431) | 0.226 (1.891, 0.665–5.378) | 0.500 (1.224, 0.680–2.203) |
| | GG | 0 | 1 | 0 | 0.704 (–) | 1.000 (–) | 0.254 (–) |
| Allele | A | 53 | 378 | 498 | 0.534 (0.729, 0.268–1.981) | 0.239 (0.553, 0.204–1.502) | 0.333 (0.759, 0.434–1.329) |
| | G | 5 | 26 | 26 | 0.534 (1.372, 0.505–3.726) | 0.239 (1.807, 0.666–4.903) | 0.333 (1.317, 0.753–2.306) |

**Notes.**
[a] Chi-square test was used to calculate the P-value.
[b] OR and 95% CI mean Odds ratio and Confidence interval, respectively.

showed a significantly low frequency in group #1 ($P = 0.033$). The frequencies of allele G were 10.34% and 4.20% in patients of group #1 and of group #3, respectively. Hence, allele G likely seemed the risk factor for patient in group #1 (Table 3).

Given that the patients in group #3 underwent convalescence, HBV DNA viral load cannot be detected in this study. The HBV viral loads of 14 acutely and 44 chronic HBV-infected individuals were tested. The results showed that HBV viral load of the acutely HBV-infected individuals was higher than that of the chronic HBV-infected group ($P = 0.0003$) (Fig. 1). The viral loads of the patients with different genotypes of each SNP were similar in groups #1 and #2.

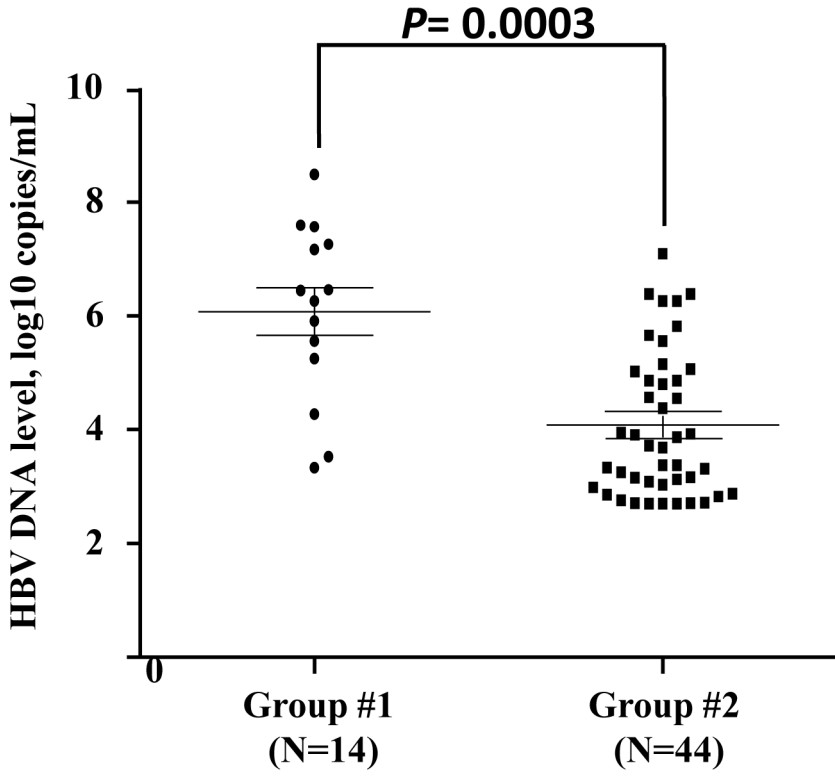

**Figure 1** **Quantification of viral load in acute or chronic HBV-infected individuals.** Each sample is represented by one circle (for acute HBV-infected individuals) or one square (for chronic HBV-infected individuals). The middle line in each group represents the mean number of the HBV viral load, together with standard error of the group.

Considering that the frequencies of genotype CC of rs12979860, GG of rs8099917, and GG of 12980275 were minor in our two cohorts (HBV-infected individuals and controls), we pooled genotypes CC and CT of rs12979860, genotypes GG and GT of rs8099917, and genotypes GG and AG of rs12980275 for further biochemical characteristics analysis (Table 4). Results showed that leukomonocyte (LYM) levels existed discrepancies between two groups of SNPs rs12979860 and rs12980275. The LYM levels of the HBV-infected individuals with genotype CC of rs12979860 ($P = 0.032$) and genotype AA of rs12980275 ($P = 0.034$) were significantly lower than that of the patients with pooled genotypes. No correlation was found between HBV viral load and LYM level.

## DISCUSSION

IL28B plays an important role in HCV infection and fateful hepatic diseases caused by HCV (*Patin et al., 2012*; *Thompson et al., 2012*). Although no association exists between HBV infection and genetic polymorphisms in the *IL28B* gene by GWAS, the *IL28B* gene could influence the viral clearance and treatment effect in different HBV-infected populations (*Stattermayer & Ferenci, 2015*). In the present study, we analyzed the genotype and allele

**Table 4  Analysis of biochemical characteristics of genotypes of each SNP in HCV-infected persons.**

| Marker | rs12979860 (Mean ± SEM) | | | Marker | rs12979860 (Mean ± SEM) | | |
|---|---|---|---|---|---|---|---|
| | CC | CT & TT | P-value[a] | | CC | CT & TT | P-value |
| AST | 38.86 ± 4.29 | 32.41 ± 4.04 | 0.598 | BUN | 5.15 ± 0.19 | 4.42 ± 0.16 | 0.168 |
| ALT | 48.16 ± 7.46 | 36.02 ± 5.22 | 0.566 | CREA | 81.78 ± 5.94 | 63.09 ± 2.08 | 0.269 |
| TBIL | 14.77 ± 0.91 | 12.85 ± 0.73 | 0.460 | UA | 344.9 ± 5.05 | 338.7 ± 13.48 | 0.683 |
| DBIL | 6.48 ± 0.71 | 4.68 ± 0.36 | 0.371 | WBC | 9.26 ± 2.47 | 7.00 ± 0.36 | 0.749 |
| IBIL | 8.28 ± 0.26 | 8.18 ± 0.44 | 0.890 | NEUT | 4.39 ± 0.13 | 4.36 ± 0.35 | 0.939 |
| TP | 72.90 ± 0.37 | 72.89 ± 0.87 | 0.993 | LYM | 1.78 ± 0.03 | 2.00 ± 0.10 | 0.032 |
| ALB | 40.54 ± 0.29 | 40.69 ± 0.70 | 0.865 | MONO | 0.42 ± 0.01 | 0.45 ± 0.03 | 0.333 |
| GLOB | 32.36 ± 0.26 | 32.20 ± 0.66 | 0.843 | EO | 0.15 ± 0.01 | 0.12 ± 0.01 | 0.255 |
| GLU | 4.90 ± 0.07 | 4.61 ± 0.10 | 0.152 | BASO | 0.03 ± 0.001 | 0.03 ± 0.003 | 0.948 |

| Marker | rs8099917 (Mean ± SEM) | | | Marker | rs8099917 (Mean ± SEM) | | |
|---|---|---|---|---|---|---|---|
| | GG & GT | TT | P-value | | GG & GT | TT | P-value |
| AST | 31.48 ± 3.92 | 38.91 ± 4.26 | 0.557 | BUN | 4.43 ± 0.16 | 5.15 ± 0.19 | 0.198 |
| ALT | 36.68 ± 5.60 | 47.98 ± 7.39 | 0.606 | CREA | 64.40 ± 2.26 | 81.46 ± 5.88 | 0.329 |
| TBIL | 13.69 ± 0.96 | 14.66 ± 0.90 | 0.719 | UA | 347.8 ± 14.26 | 343.8 ± 5.02 | 0.799 |
| DBIL | 4.93 ± 0.40 | 6.44 ± 0.70 | 0.471 | WBC | 6.89 ± 0.32 | 9.25 ± 2.45 | 0.747 |
| IBIL | 8.86 ± 0.63 | 8.21 ± 0.25 | 0.481 | NEUT | 4.31 ± 0.27 | 4.40 ± 0.13 | 0.827 |
| TP | 73.16 ± 0.90 | 72.87 ± 0.37 | 0.797 | LYM | 1.98 ± 0.10 | 1.79 ± 0.03 | 0.075 |
| ALB | 41.04 ± 0.66 | 40.50 ± 0.29 | 0.544 | MONO | 0.46 ± 0.03 | 0.42 ± 0.01 | 0.186 |
| GLOB | 32.12 ± 0.68 | 32.37 ± 0.26 | 0.760 | EO | 0.12 ± 0.02 | 0.15 ± 0.01 | 0.264 |
| GLU | 4.55 ± 0.09 | 4.90 ± 0.07 | 0.091 | BASO | 0.03 ± 0.003 | 0.03 ± 0.001 | 0.903 |

| Marker | rs12980275 (Mean ± SEM) | | | Marker | rs12980275 (Mean ± SEM) | | |
|---|---|---|---|---|---|---|---|
| | AA | AG & GG | P-value | | AA | AG & GG | P-value |
| AST | 38.92 ± 4.31 | 32.20 ± 3.89 | 0.577 | BUN | 5.16 ± 0.19 | 4.43 ± 0.16 | 0.166 |
| ALT | 48.29 ± 7.49 | 35.48 ± 5.05 | 0.539 | CREA | 81.89 ± 5.96 | 62.95 ± 1.92 | 0.255 |
| TBIL | 14.75 ± 0.92 | 13.09 ± 0.72 | 0.517 | UA | 345.5 ± 5.13 | 334.3 ± 11.71 | 0.454 |
| DBIL | 6.49 ± 0.71 | 4.70 ± 0.34 | 0.368 | WBC | 9.27 ± 2.48 | 6.97 ± 0.34 | 0.741 |
| IBIL | 8.25 ± 0.26 | 8.40 ± 0.44 | 0.847 | NEUT | 4.39 ± 0.13 | 4.34 ± 0.34 | 0.895 |
| TP | 72.85 ± 0.38 | 73.27 ± 0.83 | 0.669 | LYM | 1.78 ± 0.03 | 2.00 ± 0.10 | 0.034 |
| ALB | 40.50 ± 0.29 | 40.96 ± 0.69 | 0.585 | MONO | 0.42 ± 0.01 | 0.45 ± 0.03 | 0.384 |
| GLOB | 32.35 ± 0.27 | 32.30 ± 0.62 | 0.956 | EO | 0.15 ± 0.01 | 0.12 ± 0.01 | 0.341 |
| GLU | 4.90 ± 0.07 | 4.64 ± 0.10 | 0.194 | BASO | 0.03 ± 0.001 | 0.03 ± 0.002 | 0.859 |

**Notes.**
[a] Student t test (unpaired, two-tail) was used.

frequency of SNPs in the *IL28B* gene, district phases of HBV infection, HBV viral load, and biochemical characteristics of HBV-infected individuals.

SNPs rs12979860, rs8099917, and rs12980275 were frequently studied in HBV-infected populations (*Li et al., 2011*; *Martin-Carbonero et al., 2012*). Although analysis results were not confident, the important role of *IL28B* could not be neglected. In the present study, we did not identify the association between the genotypes and alleles of three SNPs and HBV

infection. However, the genotypes of rs8099917 were likely different between HBV-infected individuals under the acute infected phase and convalescence. This result suggested that genetic variations of the *IL28B* gene might influence HBV-infected phase. Considering that the HBV viral load was detected in only 58 HBV-infected individuals, we did not identify the association between HBV viral load and SNP genotypes in the *IL28B* gene. However, polymorphisms of the *IL28B* gene were reportedly associated with HBV viral load and liver inflammation (*Li et al., 2011*). HBV-infected sample size should be amplified for further study.

Genetic polymorphisms in the *IL28B* gene were reported not to affect HBV viral clearance in HBV-infected or HBV/HIV-coinfected persons (*Sungkanuparph et al., 2004*). Although the effect of the *IL28B* gene is unclear in HBV/HCV-coinfected persons, HBV viral load and/or viral clearance might be influenced (*De Re et al., 2016*) because of the affirmatory role of *IL28B* in HCV infection (*Ge et al., 2009*; *Thomas et al., 2009*). To exclude the effect of coinfection, we selected persons infected by HBV only and confirmed that genetic variations of the *IL28B* gene might relate to the pathogenesis of HBV infection and biochemical characteristics of HBV-infected individuals.

HBV viral load was detected in 58 HBV-infected individuals, including 14 acutely and 44 chronic infectious patients. The viral load was significantly higher in acute infectious group than in chronic group. However, no association was found between HBV viral load and genetic variations in the *IL28B* gene. Similarly, no correlation was found between HBV viral load and LYM level. These results suggested that the association between genetic polymorphisms and LYM might not be influenced by HBV viral load. Whether HBV genotype played a special role in the association needs further study.

LYM is one of the most important immune cells, which main function is to recognize and clear the bacteria, virus, and tumor. When individuals are infected by virus, LYM will be activited and induce a series of immune response. Due to the lack of analysis between LYM level and genetic polymorphisms in HBV-infected patients, we firstly performed and identified that the LYM level showed significant difference between HBV patients with differ genotypes of SNPs rs12979860 and rs12980275. Although the results needs further verification, it would help us to explore the role of LYM in HBV infection.

## CONCLUSION

No association was found between genetic variations in the *IL28B* gene and HBV infection in Yunnan population. However, LYM level could be influenced by genetic variations in the *IL28B* gene in this study. Further complex analyses should focus in studying the relationship between host genetic factors, biochemical characteristics of HBV patients, HBV infection, HBV-infected process, HBV viral load, and HBV genotypes.

## ACKNOWLEDGEMENTS

We thank all participants in this study.

### Funding

This study was supported by the National Natural Science Foundation of China (31460289) and Foundation for Innovation team of Kunming University of Science and Technology. The funders had no role in study design, data collection and analysis, decision to publish, or preparation of the manuscript.

### Grant Disclosures

The following grant information was disclosed by the authors:
National Natural Science Foundation of China: 31460289.
Foundation for Innovation team of Kunming University of Science and Technology.

### Competing Interests

The authors declare there are no competing interests.

### Author Contributions

- Yuzhu Song conceived and designed the experiments, performed the experiments, wrote the paper, prepared figures and/or tables, reviewed drafts of the paper.
- Yunsong Shen performed the experiments, prepared figures and/or tables.
- Xueshan Xia contributed reagents/materials/analysis tools, reviewed drafts of the paper.
- A-Mei Zhang conceived and designed the experiments, analyzed the data, contributed reagents/materials/analysis tools, wrote the paper, reviewed drafts of the paper.

### Human Ethics

The following information was supplied relating to ethical approvals (i.e., approving body and any reference numbers):

This study was approved by the Institutional Review Board of Kunming University of Science and Technology (Approval No. 2014SK027).

### Data Availability

The raw data is included in the Supplemental Files.

### Supplemental Information

Supplemental information for this article can be found online at http://dx.doi.org/10.7717/peerj.4149#supplemental-information.

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
