# Peer review of "Association between genetic polymorphisms of the IL28B gene and leukomonocyte in Chinese hepatitis B virus-infected individuals"

_PeerJ, doi:10.7717/peerj.4149_

## Round 0.1 · original submission · Major Revisions

I fully agree with the comments/questions/suggestions the referees have formulated. I am particularly worried about the lack of details on Material & Methods and the quality of the English language. Please be sure you correct it and satisfy our questions, checking them in the re-submission letter.

Reviewer 1 ·

Basic reporting

no comment

Experimental design

no comment

Validity of the findings

no comment

Additional comments

Dear Authors, I have read with great interest Your manuscript entitled “Association between genetic polymorphisms of the IL28B gene and leukomonocyte in Chinese hepatitis B virus-infected persons”. The study showed the interesting association between IL28B polymorphisms and HBV infection and biochemical characteristics of HBV-infected individuals with different serological expressions.
The influence of host-related genetic variability on HBV infection, HBV surface antigen seroclearance and differences in response rates to therapy in CHB patients is not well understood. Unlike many studies confirming that SNPs in the interleukin 28B (IL28B) gene play a primary role in IFN-based treatment outcomes in patients with chronic hepatitis C, the association between IL28B SNPs and the clinical course and serologic profiles of acute and chronic hepatitis B has been a subject of very few studies. Interestingly, the investigation was carried out within Chinese HBV-infected population and will be helpful for further research in this field among specific subpopulation.
It is a study with relatively large amount of cases with chronic infected phase and patients undergoing convalescence but small for patients at acute infected phase to final conclusions. Considering the above, the authors skilfully and carefully draw conclusions, point out the need for further research.
However, I have a few comments:
1. Please precise the criteria for patients undergoing convalescence in section "Subgrouping of HBV-infected persons".
2. Create a legend for the results in Table 3 (p-value, OR, 95%CI)
3. It is very interesting to compare patients who developed chronic HBV infection with patients who have undergone self-treatment – Please show the results of the comparison between the second and the third group in table 3.
4. Authors identified the relationship between IL28B genotypes and biochemical parameters, but these analyzes were performed within the entire group of patients enrolled in the study. These patients are in various phases of HBV infection (acute, chronic and convalescence) and may therefore have different biochemical parameters. Please explain why not have done these analyzes.
Please provide biochemical results in three patient groups. If no significant differences between these three groups are found in terms of biochemical parameters then table 4 may remain. On the other hand, if the patient groups differ significantly in biochemical parameters, then additionally, please analyze the relationship between IL28B and biochemical parameters for each group separately for IL28B dominant models (body article or supplemental).

Reviewer 2 ·

Basic reporting

Background on IL28B polymorphisms is all right.

Background on the role of leukomonocyte (LYM) levels in HBV patients is lacking.

There are some grammar mistakes that should be revised.

Experimental design

The design is ok as well as the technical standard.

Validity of the findings

The findings are valid, however in my opinion the interpretation of the results are not completely right.

Additional comments

The authors evaluated the role of the IL28B gene in HBV-infected individuals in China. They screened genotypes of three single nucleotide polymorphisms (SNPs, rs12979860, rs8099917, and rs12980275) in HBV infected persons and general controls by using SnapShot and sequencing. The results showed no significant differences in genotypes and alleles between the HBV infected persons and controls. Haplotype frequency was also similar between the HBV infected patients and controls. On the other hand genotype GT (P = 0.033) and allele G (P = 0.038) of rs8099917 showed statistically higher frequency in patients with acute hepatitis than compared to HBV patients undergoing convalescence. They also found that leukomonocyte (LYM) levels were associated with SNPs in the IL28B gene. In addition, the LYM levels were lower in the HBV-infected persons with genotype CC of rs12979860 and AA of rs12980275 than in the patients with other genotypes of these two SNPs. They concluded that genetic polymorphisms of the IL28B gene might be associated with the
pathogenesis or treatment effect of HBV infection.

Grammar has some errors that should be revised.
I do not think the term “popular disease” should be used. Instead, the authors should say “important disease” or “relevant disease”.
The abstract should specify the studied population (acute hepatitis B, chronic HBV and convalescence).
Major issues
I do not agree that differences in leukomonocyte (LYM) levels are striking results. If so, the authors should have reported the background on the issue. I do not see relevance on this finding.
Conclusions “genetic polymorphisms of the IL28B gene might be associated with the pathogenesis or treatment effect of HBV infection” are not supported by the results.

·

Basic reporting

English language of the manuscript is very poor, starting from the abstract, with terms and expressions that are not correct and sometimes totally wrong (i.e. Abstract background:” Hepatitis B infection is the most popular hepatic disease in China”. Most popular ”??? This is an expression used for other issues). My suggestion is to have a native English speaker perform a deep correction of the text.
The background is too superficial. The previous works are cited without a clear explanation of their findings and main messages: i.e. “However, a genome-wide association study (GWAS) among Asians found that this gene is not correlated with HBV (Kamatani Y. et al. 52 2009; Hu Z. et al. 2013).”: correlated to what? Viral clearance? Treatment outcome? HCC risk? Faster or slower evolution of progressive liver damage? Please explain. Also: “Recently, genetic variations of the IL28B gene have been identified to be associated with HCV infection, viral clearance, and response to therapy” : the first papers on this issue have been published in 2009… they are not recent.
Then further, in the abstract: “ However, the single-nucleotide polymorphisms (SNPs) or the haplotypes constructed by SNPs in the IL28B gene can influence the HBV infection, HBV surface antigen sero-clearance, or treatment of HBV-infected persons (Seto W. K. et al. 2013; Domagalski K. et al. 2014).” This appears in contrast with the sentence above, but no explanation was given.
Then, further in the background: “The present study investigated whether SNPs in the IL28B gene influence the HBV infection and biochemical characteristics of HBV-infected individuals with different serological expressions in Yunnan, China.” Reading the manuscript it is hard to understand which are the biochemical characteristics the authors are referring to. The authors reported some data about serological HBV parameters in the supplementary materials (the files are named “supplementary”, please change), but this is clearly not enough for a well designed study related to a hepatotropic virus. What about the other liver-related parameters (ALT, AST, bilirubin, albumin etc…)? My suggestion is to add a table with all the patient features, even if the same subjects have been analyzed in a previous study.
In summary, intro & background are to superficial and the literature could be better explained to understand the novelty coming from this study. Raw data are supplied but they are not enough to perform a complete and solid analysis (all the main liver parameters are completely missing). A table reporting all the patients features is mandatory.

Experimental design

The aim of the study is not completely clear as already stated. Methods are not described with sufficient detail & information; usually, even if the methods are described in a previous paper, a brief explanation of the techniques could be helpful, at least to understand the type of approach. Here this description is completely missing. Please, add more information about patients and methods and try to collect more data about the patients to make a exhaustive and maybe find some association with other liver/disease related parameters.

Validity of the findings

Although the results could raise some interest the impact and the novelty of this study, in the present version, are not sufficient . I suggest to implement the data related to patients and also perform new analyses on them. This could make the paper suitable for publication.

Additional comments

The paper needs a deep revision and many changes are needed:
-English language editing is mandatory;
- making the aim of the study clearer is also necessary;
- more laboratory and clinical data of the studied patients could open the way to more interesting analyses (multivariate analysis);
- Background should be more detailed and clearer to easily understand the novelty of this research;

Some concepts should be corrected and expressed in a more scientific way:
“HBV-infected persons (persons it is a very unusual word, please change) were also devoid of any liver disease”. It is very strange that no one patient had liver damage since the cohort is pretty wide. How did the authors assessed the liver damage?
also:” Although no consistent result is obtained in different populations, HBV infection is reportedly associated with the IL28B gene “ how can an infectious disease be related to some particular host genotype? Maybe the pathogen clearance or the higher chance to be successfully treated could be related to a particular genetic pattern. The contagion is completely independent by the host genetics.
“Subgrouping of HBV-infected persons” paragraph:
“Group 3 comprised HBV-infected persons undergoing convalescence (N = 262).” What do the authors mean with the term “convalescence”?

My suggestion is to provide all the missing infos and to make all the suggested changes in order to improve the manuscript.

---

## Round 0.2 · Minor Revisions

Please, have the minor typing mistakes corrected before to resubmit the final version of the paper.

Reviewer 1 ·

Basic reporting

Background on IL28B polymorphisms and the role of leukomonocyte (LYM) levels in HBV patients is all right.
The manuscript is at great length, well written, seems accurate and well organized and the abstract represent the content of the paper. The authors clearly presented any doubts concerning the investigation conducted by them. The manuscript is discusses the reported findings in the context of the present knowledge - literature review is adequate.

Experimental design

The aim of the study is clear. Methods are described with sufficient details.

Validity of the findings

The findings are valid. The paper makes original contribution and it is clinically exhaustive. There are some limitations regarding the small size of some subgroups; However, as the authors mentioned in the review division of HBV-infected individuals into three subgroups to analyze the biochemical characteristics, but no statistical difference was identified. The results of these analyzes should be included in the form of a table to the supplement. This could make the paper suitable for publication.

Additional comments

The study showed the interesting association between IL28B polymorphisms and HBV infection and biochemical characteristics of HBV-infected individuals with different serological expressions.
The influence of host-related genetic variability on HBV infection, HBV surface antigen seroclearance and differences in response rates to therapy in CHB patients is not well understood. Unlike many studies confirming that SNPs in the interleukin 28B (IL28B) gene play a primary role in IFN-based treatment outcomes in patients with chronic hepatitis C, the association between IL28B SNPs and the clinical course and serologic profiles of acute and chronic hepatitis B has been a subject of very few studies. Interestingly, the investigation was carried out within Chinese HBV-infected population and will be helpful for further research in this field among specific subpopulation.
It is a study with relatively large amount of cases with chronic infected phase and patients undergoing convalescence but small for patients at acute infected phase to final conclusions. Considering the above, the authors skilfully and carefully draw conclusions, point out the need for further research.
I have no comments.

Reviewer 2 ·

Basic reporting

The article meets the required standards.

Experimental design

The article meets the required standards.

Validity of the findings

The article meets the required standards.

Additional comments

The authors have made the required changes.

·

Basic reporting

I think the authors' replies to the reviewer's questions are satisfying; the authors made the required changes in the text. There are still some minor typing mistakes to correct ("Untile" instead of until... etc.).

Experimental design

The authors made the required changes

Validity of the findings

See the review of the first version

Additional comments

Please, have the minor typing mistakes corrected before to resubmit the final version of the paper.

---

## Round 0.3 · accepted · Accept

Minor mistakes present in the previous version corrected. I think the manuscript is now suitable for publication.

Reviewer 1 ·

Basic reporting

The article meets the required standards.

Experimental design

The article meets the required standards.

Validity of the findings

The article meets the required standards.

Additional comments

The authors have made the required changes.

·

Basic reporting

No new comments regarding the current revised version.

Experimental design

No comment

Validity of the findings

No comment

Additional comments

The authors fixed the minor mistakes present in the previous version. I think the manuscript is now suitable for publication.